# Surgical Outcomes of Secondary Alveolar Bone Grafting and Extensive Gingivoperiosteoplasty Performed at Mixed Dentition Stage in Unilateral Complete Cleft Lip and Palate

**DOI:** 10.3390/jcm9020576

**Published:** 2020-02-20

**Authors:** Yu-Ying Chu, Frank Chun-Shin Chang, Ting-Chen Lu, Che-Hsiung Lee, Philip Kuo-Ting Chen

**Affiliations:** 1Division of Trauma Plastic Surgery, Department of Plastic and Reconstructive Surgery, Chang Gung Memorial hospital, Linkou 333, Taiwan; yychumonica0320@gmail.com (Y.-Y.C.); ikemanleee@gmail.com (C.-H.L.); 2Craniofacial Research Center, Department of Medical Research, Department of Plastic Surgery, Chang Gung Memorial Hospital, Taoyuan 333, Taiwan; frankchang@adm.cgmh.org.tw (F.C.-S.C.); tingchenlu@gmail.com (T.-C.L.); 3Graduate Institute of Chemical and Materials Engineering, College of Engineering, Chang Gung University, Taoyuan 333, Taiwan; 4Craniofacial Center, Taipei Medical University Hospital and Department of surgery, Taipei 110, Taiwan

**Keywords:** secondary alveolar bone grafting, extensive gingivoperiosteoplasty, bone volume, perioperative orthodontic treatment, alveolar cleft repair, CT measurement of alveolar defect

## Abstract

Secondary alveolar bone grafting (SABG) is associated with donor site morbidities. We aimed to compare the outcomes of SABG and extensive gingivoperiosteoplasty (EGPP) at the mixed dentition stage. This single-blinded, randomized, prospective trial enrolled 50 consecutive patients with unilateral complete cleft lip and palate who had residual alveolar bone cleft, of which 44 (19 SABG, 25 EGPP) completed the study. Bone volumes before surgery, 6 months postoperatively, and 1-year postoperatively were compared using computed tomography. The Bergland scale score was recorded at 6 months postoperatively. Both groups had the same preoperative alveolar cleft volume. On the Bergland scale, 21, 3, and 1 patient in the EGPP group and 16, 2, and 1 patient in the SABG group were classified as types I, II, and IV, respectively, which did not show significant difference. With perioperative orthodontic treatment, the 1-year residual bone defect volume in both groups did not show significant difference (SABG 0.12 cm^3^ vs. EGPP at 0.14 cm^3^, *p* > 0.05). The study was not able to reveal much difference between SABG and EGPP combined with perioperative orthodontic treatment.

## 1. Introduction

Alveolar cleft is present in the majority of patients with complete cleft lip and palate. Alveolar bone grafting (ABG) for cleft patients restores the function and structure of the maxillary arch [1,2]. The bone graft as the third layer can better help to close the oronasal fistula if existing and improve the status of oral hygiene [3]. Secondary alveolar bone grafting (SABG) has become the standard procedure for repairing alveolar cleft in many centers. It has beneficial effects on facial growth compared with primary gingivoperiosteoplasty (GPP) [3,4]. Although SABG is the standard procedure for alveolar cleft repair, it is associated with donor site morbidities, such as pain, hematoma, and long surgical scar, though generally complications are at a very low level. Resorption and infection may also occur at the graft site. 

GPP is considered as a technique for alveolar bone rehabilitation, which stabilizes the maxillary arch and promotes spontaneous eruption of deciduous and permanent teeth by creation of the bone bridge [5]. For the reason, the debate between ABG and GPP remains controversial. 

This study aims to compare outcomes, such as complication rate and bone volume formation, of SABG and EGPP at the mixed dentition stage.

## 2. Materials and Methods

This single-blinded, randomized, prospective clinical trial was approved by the Institutional Review Board of Chang Gung Memorial Hospital (IRB 102-184A3). The study protocol complies with the principles of the Declaration of Helsinki. Patients and legal guardians of the patients provided informed consent.

### 2.1. Patients and Randomization

This study enrolled 50 consecutive patients with unilateral complete cleft lip and palate who had residual alveolar bone cleft between December 2013 and September 2016. The inclusion criteria were Taiwanese patients with nonsyndromic complete unilateral cleft lip and palate, and at the age of mixed dentition. All patients received primary cheiloplasty at the age of 3 months and primary palatoplasty at the age of 9 to 12 months. The exclusion criteria were the presence of other craniofacial anomalies and nonconsenting patients or legal guardians. All surgeries were performed at the Craniofacial Center of Chang Gung Memorial Hospital by two cleft surgeons.

Patients were block randomized to a 1:1 ratio by a third-party specialized nurse, who was not involved in the study, into group I (SABG) and group II (EGPP) using secure randomization envelopes. Randomization codes were not revealed to outcome assessors; therefore, they remained blinded throughout the imaging study. However, this was a single-blinded study; although the assessor was blinded to patient grouping, the surgeon and patients knew whether there was a bone graft; the patients knew of the bone graft because of the donor site wound. We also reviewed patients’ charts, X-rays of dentition, sex, age, operation methods, hospital stay course, and postoperative complications.

### 2.2. Surgical Procedure

The surgical technique of SABG or EGPP is the same initially, the only difference is on the presence or absence of bone grafting [6]. ABG was performed similarly to the description of Chen, Huang, and Noordhoff (Figure 1) [7]. The oral cavity was irrigated with 0.1% chlorhexidine gluconate solution. Then, 1% xylocaine with 1:200,000 epinephrine was used as local anesthesia for the upper buccal sulcus and gingiva injection. The gingival incision line was created along both sides of the alveolar cleft. A superiorly based gingival mucoperiosteal flap with its blood supply based on the superior area was elevated from the edge of the lesser segment and deepened to the first molar posteriorly. The incision line was then changed into an oblique direction, heading to the buccal sulcus. The medial segment flap was elevated in the same way, extensively to the canine of the contralateral side. Palatal mucoperiosteal flaps were dissected deep until beyond the most posterior margin of the buccoalveolar fistula. After elevating the palatal tissue to completely separate the nasal floor from the palatal mucoperiosteum, the fistula on the palatal side was securely repaired. The nasal floor tissue was dissected cranially to reach the pyriform aperture laterally and the cartilage of the nasal septum on the medial side. The nostril sill vertical discrepancy can be corrected well with a periosteal tension-free closure of the nasal floor.

For the SABG group, the iliac crest was prepared as the donor site of the cancellous bone. The bony defect was packed with bone chips from the alveolar process to the pyriform aperture. The periosteum of the lateral gingival flap was scored to reduce the tension at the lateral end of the incision. This allowed a tension-free advancement of this lateral gingival flap. It was sutured to the palatal and medial flaps to achieve watertight closure without tension [8].

Some patients in both groups received perioperative orthodontic treatment to align the teeth for correction of incisor inclination or rotation. Orthodontic treatments were started 6 months preoperatively by orthodontic specialists and continued for 6 months postoperatively [8].

### 2.3. Bergland Scale Score

The Bergland scale score [9] was assessed using plain two-dimensional dental radiographs. The Bergland scale score was obtained based on the following criteria: type I, bone height is approximately normal; type II, bone height is at least three-quarters of the normal bone height; type III, bone height is less than three-quarters of normal bone height; type IV, absence of a continuous bone bridge across the cleft gap. Types I and II are defined as clinical success and types III and IV as clinical failure.

### 2.4. Measurements on Three-Dimensional Computed Tomography (3D-CT)

CT was performed preoperatively, 6 months postoperatively, and 1 year postoperatively with a slice thickness of 1 mm. Data were processed by Avizo 7.0 software (Visage Imaging, Carlsbad, CA, USA) in Digital Imaging and Communications in Medicine format. The Frankfort plane was used for the orientation of the cranium in the sagittal, coronal, and axial views. Multiplanar imaging was used to define the bone defect, bone graft area, and medial incisor. Alveolar bone defect, bone grafting area, and residual defect volume were assessed by 3D-CT as the primary outcome [8].

### 2.5. Statistical Analysis

Patients’ sex, cleft affected side, orthodontic treatment or not, canine eruption, and Bergland scale score were compared with the Chi-squared test. Patients’ age and bony defect change rate were compared with the Mann–Whitney U test. Patients’ preoperative and postoperative bony defects were compared with a Wilcoxon signed rank test. Statistical analysis was processed with IBM SPSS 18.0 (IBM Corp, Armonk, NY, USA) with significance set as *p* < 0.05.

## 3. Results

### 3.1. Demographic Data

Of the 50 patients initially identified, 44 completed the study, which included 19 patients in the SABG group and 25 patients in the EGPP group. The average age when surgery was performed was 9.58 (range, 8.54 to 12.3) years. Demographic data are shown in Table 1. Of the 44 patients, 22 (50%) have perioperative orthodontic treatment, with 11 patients in each group.

### 3.2. Preoperative Comparison

The mean preoperative bony defect in the SABG and EGPP groups was 0.50 cm^3^ and 0.49 cm^3^, respectively. No significant difference was found between the two groups preoperatively. We further divided the patients into four subgroups according to whether they received perioperative orthodontic treatment or not (Table 2). In the SABG group, the preoperative bony defect volume in 11 patients with orthodontic treatment was 0.55 cm^3^. In the EGPP group, the preoperative bony defect volume in 11 patients with orthodontic treatment was 0.49 cm^3^. No significant difference was found.

### 3.3. Postoperative Comparison

#### 3.3.1. Bergland Score

On the Bergland scale, 21, 3, and 1 patient in the EGPP group and 16, 2, and 1 patient in the SABG group were classified as types I, II, and IV, respectively, which did not show significant differences (Table 3). However, care should be taken as one patient in each group presented no bony bridge growing across the cleft gap, which was classified as type IV, but no patient had type III. 

#### 3.3.2. CT Measurement

At 1-year follow-up, the residual bony defect volume in the SABG group and EGPP group was 0.12 cm^3^ and 0.23 cm^3^, respectively, which showed significant differences between the two groups. However, we divided the SABG and EGPP groups into two subgroups depending on whether they have perioperative orthodontic treatment or not. The comparison of both subgroups with perioperative orthodontic treatment is shown in Table 3. The bony defect at 6-month follow-up was 0.16 cm^3^ in the SABG orthodontic treatment subgroup and 0.21 cm^3^ in the EGPP orthodontic subgroup; and at 1-year follow-up was 0.12 cm^3^ and 0.14 cm^3^, respectively. Those results of residual bony defect volume showed no statistically significant difference between both subgroups of which the patient received perioperative orthodontic treatment. 

#### 3.3.3. Complication

After the surgery, one patient in the SABG group presented wound infection with foul odor and turbid discharge. This patient received two weeks of antibiotics, and the infection subsided finally. No complication was seen in the EGPP group. The complication rate did not show significant difference. 

#### 3.3.4. Canine Eruption

At 1-year follow-up, one patient showed canine impaction, whereas the other 18 patients in the SABG group had canine eruption. Compared with the EGPP group, one patient had canine impaction and the other 24 patients had canine eruption (Table 4). The results did not show significant differences between the two groups. 

## 4. Discussion

Although SABG performed with autogenous bone is the elementary part of the surgical protocol in the majority of centers all over the world because of its efficiency, this study aimed to evaluate the efficiency of the method which eliminates donor site necessity. We compared the outcomes, such as complication rate and bone volume formation, of SABG and EGPP performed at the mixed dentition stage. 

ABG as a standard procedure in alveolar cleft is generally classified into primary and secondary. Secondary ABG is usually performed in 9- to 11-year-old patients, in which cancellous bone is harvested from the iliac crest and grafted into the alveolar cleft before canine teeth eruption [10]. The surgical timing of our patients was also at mixed dentition age. 

ABG is undoubtedly recognized as an effective method to promote more bony stock for the maxillary teeth. Nonetheless, the most discussed drawback of ABG is its permanent scar and donor site morbidity. According to Bykowski et al., patients complained of pain after bone graft harvesting, which usually reached a pain score of 7.3/10, in the open harvest group. The pain of donor site was worse than pain associated with cleft reconstruction intraorally. It also prolongs the hospital length of stay [11].

The other reported complications are hematoma, revision surgery for bleeding, infection, delayed wound healing, and sensory disturbances of the lateral femoral cutaneous nerve [12]. Given the complications of ABG, surgeons are searching for alternative methods to replace or enhance autogenous bone graft.

Periosteoplasty was first described as a “boneless bone grafting” by Skoog in 1967 [13]. It is performed to remove the soft tissue barrier in the cleft and create a mucoperiosteal tunnel, which can restore the continuity of the attached gingiva, maintaining the space for bone regeneration. Skoog recommends performing periosteoplasty in infancy, and good bone formation of the alveolar cleft was noted in 47% of patients [14]. 

Some centers perform primary GPP with a high successful rate [15]. In 2012, Chou et al. reported 68.2% successful bone formation rate in primary GPP in patients aged 3 months [16]. Compared with delayed periosteoplasty, which is performed between infancy and before 7 years of age, good bone formation was found in 80% of patients [14]. As for early secondary gingivoalveoloplasty, Meazzini reported that 71.7% of patients developed type I alveolar bridging by modified Bergland scale score [17].

Although primary GPP has fair results in bone regeneration, it was found to affect the sagittal growth of the maxilla at 5 years of age [4]. Early secondary gingivoalveoloplasty also appeared to have an inhibiting effect on maxillary growth that may increase the need for further orthognathic surgery [18]. 

According to Brudnicki et al., not only GPP but also ABG surgery, regardless of the timing of surgery, can cause vertical and horizontal maxillary growth restriction [19]. Compared with the late surgery group, the early ABG group presented more severe growth disturbance in both the sagittal and vertical directions [20]. These results indicate that facial growth disturbance relates to early alveolar surgery, explaining the fact that both GPP and ABG affect the vertical height of the maxilla.

Considering the benefit of tooth eruption, the current trend is to perform surgery at the age of mixed dentition [9,21,22,23]. Dental age is our main consideration for determining the timing of surgery.

According to our results, after perioperative orthodontic treatment, the 1-year volume change in the EGPP group reached 69.8%, which was not significantly different compared with that in the SABG group (79.4%). The Bergland scale assessment revealed that most patients in both groups were classified as type I; the canine eruption rate was also not significantly different in both groups. These results were compatible with the CT image study. Our data proved that combined with orthodontic treatment, the bone formation in the EGPP group was similar to that in bone grafting. 

The osteogenesis phenomenon after orthodontic treatment is described in the literature. Meazzini et al. found that early secondary gingivoalveoloplasty promotes adequate bone growth in the alveolar and nasal regions [24]. The rate of permanent tooth eruption remains normal [18]. Nováčková et al. presented that the bone created via orthodontic treatment appeared to have sufficient stability in horizontal and vertical directions at long-term follow-up [25]. The biophysical mechanism is the regulation of bone remodeling by signaling factors. Tooth loading during orthodontic treatment causes hypoxia, which initiates an inflammatory cascade, inducing osteoblast deposition in the tension area and osteoclast resorption in the compression area [26]. 

The idea of EGPP for the alveolar cleft instead of bone grafting came from the experience in some patients in whom alveolar fistula repair was difficult, such as very long and deep fistulas and wide alveolar gaps in syndromic patients that cannot cooperate with perioperative orthodontic treatment, and graft infection due to poor oral hygiene. Instead of SABG, EGPP was performed in these patients to prevent graft infection from difficult wound closure. During follow-up, bone formation was found in most of these patients after the EGPP.

Another crucial point is how wide the dissection area is in EGPP. Without bone grafting, the nasal floor, as the roof of the cleft, may collapse and contact with the palatal mucosa floor, as the redundant soft tissue causes obliteration of the cleft space, allowing less bone growth. EGPP enables deep dissection to the most posterior part of the cleft and excision of redundant soft tissues as much as possible. The tensile force of the nasal floor and palatal mucosa become high, preventing the collapse of the cleft space. Hence, the bone is allowed to regenerate within the space during orthodontic treatment without impediment by soft tissue. If the dissection in GPP is not extensive enough or if the soft tissue is not trimmed adequately, alveolar bone growth is compromised. 

Another alternative method to replace or enhance autogenous bone graft is recombinant human bone morphogenetic protein-2 (rh-BMP2) [27]. From a recent meta-analysis, the rh-BMP2 graft showed comparable effectiveness to autologous bone in terms of bone graft volume and height, but provides a shorter hospital stay due to the absence of a donor site [28]. EGPP also reduces hospital stay, and in this study, no complications after EGPP were reported. 

The limitations of this study were the small sample size and short follow-up period. The method error assessment was not conducted. Further study with larger sample size and long-term follow-up is needed. 

## 5. Conclusions

The results of this study were not able to reveal much difference between EGPP combined with perioperative orthodontic treatment and the SABG group. With the additional benefit of elimination of the donor site, EGPP is considered a method for treatment of alveolar cleft that may continue in the future. 

## Figures and Tables

**Figure 1 jcm-09-00576-f001:**
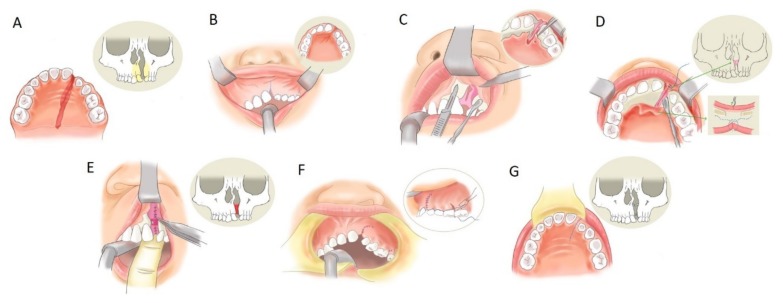
Extensive gingivoperiosteoplasty. Alveolar cleft and bone defect (**A**). Incision line made on the upper gingiva along with the cleft (**B**). The gingival mucoperiosteal flap, in which the blood supply comes from the superior part, was elevated to explore the cleft (**C**). The palatal mucoperiosteal flap deep to the margin of the alveolar fistula was elevated (**D**). Then, the tissue of the nasal floor was watertight sutured (**E**). The bone graft was packed into the cleft area at this step (secondary alveolar bone grafting (SABG) method). The gingival site and palatal site wound were closed completely (**F,G**).

**Table 1 jcm-09-00576-t001:** Demographic data.

		SABG	EGPP	*p*-Value
total no.		19	25	
sex	female	4	12	0.066
	male	15	13	
orthodontic treatment	yes	11	11	0.361
	no	8	14	
affected side	right	6	12	0.272
	left	13	13	
age		9.55	9.6	0.310

EGPP, extensive gingivoperiosteoplasty; SABG, secondary alveolar bone grafting.

**Table 2 jcm-09-00576-t002:** CT volume measurement of subgroups with perioperative orthodontic treatment.

	SABG	EGPP	*p*-Value
perioperative orthodontic treatment			
number	11	11	
volume of bony defect (cm^3^) (mean (SD))			
preoperative	0.551 (0.194)	0.494 (0.186)	0.493
POM6	0.162 (0.116)	0.208 (0.111)	0.392
POY1	0.115 (0.116)	0.141 (0.071)	0.528

POM 6, postoperative month 6; POY1, postoperative year 1; SABG, secondary alveolar bone grafting; CT, computed tomography; EGPP, extensive gingivoperiosteoplasty.

**Table 3 jcm-09-00576-t003:** Bergland scale score.

Type	I	II	III	IV		
					Total	*p* Value
SABG	16	2	0	1	19	
EGPP	21	3	0	1	25	
	37	5	0	2	44	0.549

SABG, secondary alveolar bone grafting; EGPP, extensive gingivoperiosteoplasty.

**Table 4 jcm-09-00576-t004:** Canine eruption.

Canine Status	Impaction	Eruption		
			Total	*p* Value
SABG	1	18	19	
EGPP	1	24	25	
	2	42	44	0.84

SABG, secondary alveolar bone grafting; GPP, extensive gingivoperiosteoplasty.

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
