# Peer review of "Surgical Outcomes of Secondary Alveolar Bone Grafting and Extensive Gingivoperiosteoplasty Performed at Mixed Dentition Stage in Unilateral Complete Cleft Lip and Palate"

_jcm, 2020, doi:10.3390/jcm9020576_

Round 1

Reviewer 1 Report

14-February-2020

Title: Surgical Outcomes of Secondary Alveolar Bone Grafting and Extensive Gingivoperiosteoplasty Performed at Mixed Dentition Stage in Unilateral Complete Cleft Lip and Palate

Comments and Suggestions for the Authors

The authors of this study prospectively analyzed and compared the surgical outcomes of SABG and gingivoperiosteoplasty at mixed dentition stage in non-syndromic patients suffering from UCLP.

The study evaluated the interesting topic, currently investigated and discussed in the field of cleft surgery.

Although the study was well designed and ethically approved (despite repeated and not mentioned in the text levels of exposition to x-rays) and included 3D volumetric methodology in estimation of obtained results, the credibility of their conclusion was severely undermined by both inappropriate numbers of compared subgroups and misinterpretation of the obtained results. What’s more the writing style is narrative and includes much opinion and discussion not specifically arising from the results. Major flaws are as follows:

The text needs to be improved and rewritten – at the moment it is inconsistent leaving a wide space for interpretation and assuming ‘what the authors meant’. The authors subjectively tend to support gingivoperiosteoplasty as equal to SABG even against own results – see lines 175 and 176 - which reveal the most important results of the study, and not even mentioned in the abstract. Instead of this, the authors concentrate too much on the subject of perioperative orthodontic treatment which was not the aim of this study and at the same time can be misleading in interpretation of the results – see table 3; see line 195 and 196. The presented study model doesn’t allow authors to conclude about enhancement of bony growth by orthodontic treatment. The ‘no subgroups’ in table 3 consist of 8 and 4 patients! For the sake of the clarity of this manuscript, the exploration of orthodontic treatment in this study should be limited to mentioning the similar percentage of orthodontically treated patients in compared groups – SABG and EGPP (which is well done in table 1). The study is devoted to the subject requiring high numbers of patients in compered groups because of the very subtle differences of expected results. So, the interpretation of the results obtained in this study should be very cautious and should sound like ‘the study was not able to reveal much difference between SABG and EGPP groups’ instead of what is mentioned in lines 31, 32 – ‘With perioperative orthodontic treatment, both groups can achieve similar results. EGPP may be an alternative for alveolar cleft repair …’ Moreover, the study doesn't describe the method error assessment. The avoidance of a scar in invisible area for the cost of the risk of compromised results of alveolar repair with very visible consequences doesn’t sound convincing. Do the authors think that somebody with UCLP and more or less disfigured face after 5 or 7 surgical operations cares about a scar or irregularity in the region of a hip? – line 212 and 213 ‘bone graft harvesting may affect the shape of the iliac bone, interfering patients’ self-image in addition to the facial scar.’

Abstract:

It should state the numbers of patients in compared groups – SABG and EGPP, otherwise is misleading.

The last sentence is inappropriate and has to be deleted or undergo major change.

Introduction:

The Introduction would benefit from rewriting in some parts as follows.

Line 40 – ‘This could also close the oronasal fistula’ – should be rather: the bone graft as the 3rd layer can help more to close the oronasal fistula if existing.

Line 40 – ‘and decrease the rate of oronasal regurgitation and nasal 40 allergy.’ - here the references should be given.

Lines 43, 44 – it would be appropriate to mention that generally donor site complications are at very low level, maybe some reference regarding donor site morbidity at different age would add some interesting facts to the Introduction.

Line 44 – figure 1 is pointless and should be deleted – everybody has already seen a scar.

Line 44 – starting from ‘Resorption and infection … ‘ till line 51. This text is interesting and should be more coherent and clear to the reader. It is not a poem so nobody wonts to assume what the author really meant.

Lines 52 and 53 - has to be changed – donor site morbidity doesn’t mean that SABG requires alternative method!

Material and Methods:

It would be appropriate to mention the surgical protocol or at least previous primary and secondary surgeries in the material since these procedures can influence alveolar reconstruction results.

3D methodology should be explained in more detail and maybe a relevant figure illustrating this methodology would give more informative value to this manuscript.

Results

Table 1 – the first line should include ‘p value’ while Operation method should be deleted – SABG and EGPP are the study groups!

Table 2 – at the moment, illustrates that there is a different before and after procedure - so what? It is trivial. This table should rather illustrate what is the most important in this study - the difference in outcome between the groups. So the first line in table 2 should include ‘SABG’ and ‘EGPP’ and 'p value' while the first column should include ‘preoperative’ and ‘POY1’.

Table 3 – should be deleted or completely rewritten – EGPP group includes 15 patients? – p value comes from comparison of what? – 'yes SABG' and 'yes EGPP'? There are many options of interpretation of this table - now it is completely unclear.

Figure 3 looks as if it is relevant but in fact gives no further information to this manuscript. The authors should seriously consider deleting it.

Discussion:

Generally should be more coherent and needs to be rewritten and developed to become more interesting and informative to the reader.

And maybe it should include the statement like: although SABG performed with autogenous bone, is the elementary part of the surgical protocol in the majority of centers all over the world because of its efficiency; this study was aimed to evaluate the efficiency of the method which eliminates donor site necessity.

Lines 194, 195 need to rewritten or deleted as mentioned previously.

Lines between 197 and 204 – the text must be coherent or at least logically continued.

Line 209 – ‘… affects overall satisfaction of the surgery,’ that is speculation if not supported by references.

Line 2010 – ‘… these patients, aged 9–12 years, have mature mental health to overcome this complication.’ Again, that is speculation if not supported by references.

Line 212 – ‘bone graft harvesting may affect the shape of the iliac bone, interfering patients’ self-image – again, very speculative - generally unprofessional text.

Line 213 – ‘The scar becomes longer proportionally as the patient grows (Figure 1)’ – trivial and adds nothing new to the discussion.

Line 276 – The limitations of the study – this must be developed and reformulated accordingly. Two surgeons is not a problem if both of them performed both kings of the procedures evaluated in the study (this information should be included in the manuscript for instance in table 1.)

The authors may claim to continue this method because they think it is promising or claim that they discontinued because the results appeared to be unsatisfactory.

To conclude:

To meet the standard of scientific journal the authors of this article need to add a lot of additional work to the manuscript and add some more information before it can be considered for publication.

Reviewer 2 Report

see attachment
